# Improved Utility Analysis of Private CountSketch

**Rasmus Pagh**
Basic Algorithms Research Copenhagen
University of Copenhagen
pagh@di.ku.dk

**Mikkel Thorup**
Basic Algorithms Research Copenhagen
University of Copenhagen
mikkel2thorup@gmail.com

## Abstract

Sketching is an important tool for dealing with high-dimensional vectors that are sparse (or well-approximated by a sparse vector), especially useful in distributed, parallel, and streaming settings. It is known that sketches can be made differentially private by adding noise according to the sensitivity of the sketch, and this has been used in private analytics and federated learning settings. The post-processing property of differential privacy implies that *all* estimates computed from the sketch can be released within the given privacy budget.

In this paper we consider the classical CountSketch, made differentially private with the Gaussian mechanism, and give an improved analysis of its estimation error. Perhaps surprisingly, the privacy-utility trade-off is essentially the best one could hope for, independent of the number of repetitions in CountSketch: The error is almost identical to the error from non-private CountSketch plus the noise needed to make the vector private in the original, high-dimensional domain.

## 1 Introduction

*CountSketch* was introduced by Charikar, Chen, and Farach-Colton (2004) as a method for finding heavy hitters in data streams. In machine learning the same sketch was studied by Weinberger, Dasgupta, Langford, Smola, and Attenberg (2009) under the name *feature hashing*, or less formally the "hashing trick". The idea is to map a high-dimensional vector $x \in \mathbf{R}^d$ to a lower-dimensional representation $Ax \in \mathbf{R}^D$, using a certain random linear mapping $A$. From $Ax$ it is possible to estimate entries $x_i$, with error that depends on $D$ and the norm of $x$, and more generally to estimate inner products. It was shown in (Weinberger et al., 2009) that for $y \in \mathbf{R}^d$, $\langle Ax, Ay \rangle$ is well-concentrated around $\langle x, y \rangle$. Recent interest in sketching techniques is motivated by distributed applications such as *federated learning* (Kairouz et al., 2021), where many users contribute to creating a model on their combined data, without transferring data itself. In this context, sketching can be used to identify the large-magnitude entries in the sum of many high-dimensional vectors (one held by each user), using communication proportional to the number of such entries.

Perhaps the most basic property of CountSketch/feature hashing is the error of estimators for a single coordinate $x_i$. The sketch directly provides independent, noisy estimators $X_1, \ldots, X_k$, each symmetric with mean $x_i$. Two main ways of combining these into a more reliable estimator for $x_i$ have been studied: Taking the *median* estimator (Charikar et al., 2004), or taking the *average* of estimators (implicit in Weinberger et al. (2009)). Minton and Price (2014) showed that using the median estimator with $k = \Theta(\log d)$ is not only is more robust, but alse decreases the variance of the estimator by a factor of $\Omega(\log d)$. Later Larsen, Pagh, and Tĕtek (2021) showed that using the median estimator with $k = 3$ makes the error decrease quadratically with the sketch size $D$.

In this paper we consider CountSketches that have been made differentially private using the *Gaussian mechanism*, which is perhaps the most widely used mechanism for making high-dimensional vectors differentially private. This sketch, which we will refer to as *Private CountSketch*, works by

36th Conference on Neural Information Processing Systems (NeurIPS 2022).

adding i.i.d. Gaussian noise to each coordinate of the CountSketch, with variance scaled according to the (squared $L_2$) sensitivity $k$ of the sketch.

We give additional evidence that the median estimator is preferable to the mean by showing that its estimation noise is independent of the number $k$ of estimators. Previous applications of sketching techniques to differential privacy have had noise that grew (polynomially) with $k$. The failure probability of CountSketch decreases exponentially in $k$, so it is preferable to have $k$ relatively large, and thus these methods faced a trade-off between the noise from the Gaussian mechanism and the error probability of the underlying CountSketch. We show that this trade-off is not needed: It is possible to *both* get a highly reliable CountSketch (by choosing $k$ large enough) *and* achieve noise that depends only on the privacy parameters (independent of $k$).

## 2   Related work

The analysis of the Gaussian mechanism is attributed in Dwork and Roth (2014, Appendix A) to the the authors of the seminal paper on differential privacy (Dwork, McSherry, Nissim, and Smith, 2006). It has since been shown to have desirable properties related to keeping track of privacy loss under composition, see e.g. Bun & Steinke (2016); Mironov (2017).

**Local differential privacy.**   A variant of CountSketch, the *Count Mean Sketch* has been used by Apple to privately collect information on heavy hitters, e.g. popular emoji in different countries (Apple Differential Privacy Team, 2017). This can be phrased in terms of summing $n$ user vectors, each one a 1-hot vector with a single 1, and identifying the large entries. Their protocol works in the *local* model of differential privacy, which means that each report is differentially private. Lower bounds on local differential privacy (Chan, Shi, and Song, 2012) imply that the noise on estimates in this setting grow with $\Omega(\sqrt{n})$, where $n$ is the number of users.

Independently, Bassily, Nissim, Stemmer, and Thakurta (2020) improved theoretical results of Bassily and Smith (2015) on heavy hitter estimation in the local model, providing practical methods for matching the lower bound of (Chan et al., 2012). They also base their method on CountSketch, though it is made private using a sampling technique combined with randomized response, not by adding noise to each coordinate. The noise from this step, rather than from CountSketch itself, dominates the error. Acharya, Sun, and Zhang (2019) showed that similar error can be achieved without any agreed-upon public randomness (with no need for a common, random sketch matrix).

Huang, Qiu, Yi, and Cormode (2022) used CountSketch, made private with the geometric (aka. discrete Laplace) mechanism, to design protocols for frequency estimation under local differential privacy and multiparty differential privacy. They observe that the estimator in each repetition of CountSketch is symmetric (assuming fully random hashing), but unlike the present paper they do not demonstrate that the median estimator has noise that is *smaller* than the noise added to each estimator. Thus their estimation error bound grows with the number of repetitions of CountSketch.

Zhou, Wang, Chan, Fanti, and Shi (2022) recently used CountSketch as the basis for a mechanism that releases $t$-sparse vectors with differential privacy. We relate our result to theirs in Section 3.4.

Independent of our work, Zhao, Qiao, Redberg, Agrawal, Abbadi, and Wang (2022) recently presented a comprehensive study of differentially private linear sketches, including Private CountSketch. They focus on bounding the maximum error, but also have theoretical results for point estimates that are weaker than ours. Their experiments confirm the performance of Private CountSketch in practice. In addition they show how CountSketch can be used to build a sketch for quantile approximation, and our results imply tighter analysis for that application.

**Protocols based on cryptography.**   Motivated by privacy-preserving aggregation of distributed statistics, Melis, Danezis, and Cristofaro (2016) considered CountSketch (and the related Count-Min sketch) made $\varepsilon$-differentially private using the *Laplace mechanism* (Dwork et al., 2016). They empirically showed that the set of *heavy hitters*, i.e., large entries in the vector, could be identified with very little error on skewed distributions, but did not provide general bounds on estimation error. Their protocol can be implemented in a distributed setting in which each user has a 1-hot vector, and we want to sketch the sum of these vectors, using cryptographic protocols for secure aggregation. This bypasses lower bounds on local differential privacy, and yields much better privacy-utility trade-

offs (but now assuming security of the aggregation protocols, so guarantees rely on cryptographic assumptions and are not information-theoretic).

Another model of differential privacy, built on the cryptographic primitive of an *anonymous channel* (e.g. implemented as a *mixnet*), is the *shuffle model* (Cheu, Smith, Ullman, Zeber, and Zhilyaev, 2019; Erlingsson, Feldman, Mironov, Raghunathan, Talwar, and Thakurta, 2019). Ghazi, Golowich, Kumar, Pagh, and Velingker (2021) used Count-Min sketches in a protocol for frequency estimation and heavy hitters in this context. Our work suggests that better accuracy can be obtained by using CountSketch instead of Count-Min, especially for high-probability bounds.

**Related results in the central model.** Mir, Muthukrishnan, Nikolov, and Wright (2011) studied general mechanisms for making sketching algorithms differentially private, and in particular studied using the Count-Min sketch to identify heavy hitters. They observed that any mechanism that makes a linear sketch private by adding (oblivious) noise to the sketch vector implies a *pan-private* sketch that can be updated dynamically. Our work implies an improved pan-private sketch based on CountSketch.

Aumüller, Lebeda, and Pagh (2021) studied differentially private representations of sparse vectors, showing that it is possible to achieve space close to the number of non-zero entries while keeping noise comparable to a naïve application of the Laplace mechanism to the raw vectors. Our work implies that, up to a logarithmic factor in space, such a result is possible with a linear sketch, which enjoys many desirable properties.

**Analysis of CountSketch.** Several authors have worked on improved analysis of the error of CountSketch in the non-private setting, including Minton and Price (2014) and Larsen, Pagh, and Tětek (2021). It seems possible to analyze Private CountSketch using the framework of (Minton & Price, 2014), but we will pursue an elementary, direct analysis that does not depend on the Fourier transform of random variables.

## 3 Private CountSketch

In this section we provide the necessary background information on CountSketch, and present our improved analysis of Private CountSketch.

### 3.1 Notation and background

We use $[k]$ to denote the set $\{1, \ldots, k\}$. Vectors are indexed by one or more integers, each from designated ranges. For example, a vector of dimension $D = kb$ may be indexed by $(i, j)$ where $i \in [k]$ and $j \in [b]$. The ordering of vector entries is not important. For integer $k$, let $\text{tail}_k(x)$ denote the vector that is identical to $x$ except that the $k$ coordinates of largest magnitude are replaced with zeros. For a predicate $P$, we let $[P]$ denote the indicator that is 1 if $P$ is true and 0 otherwise.

CountSketch is a random linear mapping of a vector $x \in \mathbf{R}^d$ to $Ax \in \mathbf{R}^D$. For suitable parameters $k$, $b$ such that $D = kb$, the sketch $CS(x)$ is defined in terms of two sequences of random, independent hash functions:

- $h_1, \ldots, h_k : [d] \to [b]$, and
- $s_1, \ldots, s_k : [d] \to \{-1, +1\}$.

CountSketch was originally presented with hash functions from a 2-wise independent families (Charikar et al., 2004), but in this paper we assume that all hash functions used are fully random. (Full randomness is used in our analysis of error, but the privacy of our method does not depend on this assumption.) To simplify our exposition we will further assume that $k$ is odd and that $b$ is even.

If we index $CS(x) \in \mathbf{R}^D$ by $(i, j) \in [k] \times [b]$, then

$$CS(x)_{ij} = \sum_{\ell \in [d]} s_i(\ell) x_\ell [h_i(\ell) = j] \ .$$

That is, each vector entry $x_\ell$ is added, with sign $s_i(\ell)$, to entries indexed by $(i, h_i(\ell))$, for $i \in [k]$. We can see the sketch as a sequence of $k$ hash tables, each of size $b$; thus we refer to $k$ as the number

of *repetitions* and to $b$ as the *table size*. It is easy to see that

$$X_i = s_i(\ell)CS(x)_{i,h_i(\ell)}$$

is an unbiased estimator for $x_\ell$ for each $i \in [k]$. Furthermore, since $s$ is fully random the distribution of $X_i$ conditioned on $s_i(\ell) = -1$ is identical to the distribution of $X_i$ conditioned on $s_i(\ell) = 1$, which means that $X_i$ is symmetric around $x_\ell$. We can combine these estimators into a more robust estimator:

$$\hat{x}_\ell = \text{median}\left(\{s_i(\ell)CS(x)_{i,h_i(\ell)} \mid i \in [k]\}\right) \quad . \tag{1}$$

Minton & Price (2014) bounded the error of $\hat{x}_\ell$, with a failure probability that is exponentially decreasing with $k$:

**Theorem 3.1** (Minton & Price (2014))**.** *For every $\alpha \in [0,1]$ and every $\ell \in [d]$, the estimation error of CountSketch with $k$ repetitions and table size $b$ satisfies*

$$\Pr\left[|\hat{x}_\ell - x_\ell| > \alpha \, \Delta\right] < 2\exp\left(-\Omega\left(\alpha^2 k\right)\right) \quad ,$$

*where $\Delta = ||tail_b(x)||_2/\sqrt{b}$.*

Note that $||\text{tail}_b(x)||_2 \leq ||x||_2$, but can be much smaller for skewed distributions and even zero for sparse vectors. Choosing suitable $\alpha^2 k = \Theta(\log d)$ we get that *all* coordinates $x_\ell$ are estimated within this error bound with high probability.

## 3.2 Private CountSketch

Several approaches for releasing a CountSketch with differential privacy have been studied. These all require that the sketches of *neighboring* vectors $x, x'$, denoted $x \sim x'$, have similar distributions. The neighboring relation that we consider is that $x \sim x'$ if and only if these vectors differ in at most one entry, by at most 1, or equivalently that $x - x'$ has a single nonzero entry with a value in $[-1, +1]$. Since CountSketch is linear, $CS(x) - CS(x') = CS(x - x')$. From this it is easy to see that for $x \sim x'$, $||CS(x) - CS(x')||_2 \leq \sqrt{k}$. In differential privacy terminology, the *sensitivity* of the sketch is $\sqrt{k}$.

There are many ways to compute a differentially private version of a CountSketch $CS(x)$ (or any other function of $x$), but it is most common to consider *oblivious* methods that work by sampling a symmetric noise vector $\nu \in \mathbf{R}^D$ that is independent of $CS(x)$ and releasing the private Count-Sketch:

$$PCS(x) = CS(x) + \nu \quad .$$

Since the noise is symmetric around zero, we have (similar to before) that $X_i = s_i(\ell)PCS(x)_{i,h_i(\ell)}$ is an unbiased estimator for $x_\ell$, and therefore it makes sense to use the median estimator (1) also for Private CountSketch:

$$\tilde{x}_\ell = \text{median}\left(\{s_i(\ell)PCS(x)_{i,h_i(\ell)} \mid i \in [k]\}\right) \quad . \tag{2}$$

Note that except for the addition of $\nu$, which can be considered an alternative initialization step, Private CountSketch works in *exactly the same way* as CountSketch: Updates are done in the same way, and estimators $\bar{x}_\ell$ are computed in the same way.

**Properties of Private CountSketch.** Since $PCS(x)$ is an affine transformation, it can be maintained under updates to $x$. Also, it is possible to add and subtract Private CountSketches (based on the same hash functions), e.g.:

$$PCS(x) + PCS(y) = CS(x + y) + \nu_1 + \nu_2,$$

where $\nu_1$ and $\nu_2$ are sampled according to the noise distribution.

The noise distribution we will consider in this paper is the standard $D$-dimensional *Gaussian* distribution $\mathcal{N}(0, \sigma^2)^D$ with mean zero and variance $\sigma^2$, and in the rest of this paper we refer to $PSC(x)$ with noise $\nu \sim \mathcal{N}(0, \sigma^2)^D$. Differential privacy properties of $PCS(x)$ follows from general results on the Gaussian mechanism, see e.g. Dwork & Roth (2014, Apendix A) and Bun & Steinke (2016). We state some of them for convenience here:

**Lemma 3.2.** $PCS(x)$ *is* $(\varepsilon, \delta)$-*differentially private for* $\varepsilon$, $\delta$ *satisfying* $\sigma^2 > 2k \ln(1.25/\delta)/\varepsilon^2$ *and* $\varepsilon < 1$, *and it is* $(k/2\sigma^2)$-*zero-concentrated differentially private.*

This tells us that to get good privacy parameters, we need to use Gaussian noise with variance $\Omega(k)$, i.e., standard deviation $\sigma$ at least $\Omega(\sqrt{k})$. The median estimator (2) returns, for some $i$,

$$s_i(\ell) PCS(x)_{i, h_i(\ell)} = s_i(\ell) \left( CS(x)_{i, h_i(\ell)} + \nu_{i, h_i(\ell)} \right)$$

i.e., a value that could be returned by CountSketch plus a noise term with standard deviation $\Omega(\sqrt{k})$. Thus, we might expect the magnitude of the noise to scale proportionally to $\sqrt{k}$. Our main result is that this does *not* happen, and in fact the magnitude of the noise can be bounded independently of $k$.

**Theorem 3.3.** *For every* $\alpha \in [0, 1]$ *and every* $\ell \in [d]$, *the estimation error of Private CountSketch with $k$ repetitions, table size $b$, and noise from $\mathcal{N}(0, \sigma^2)^{kb}$ satisfies*

$$\Pr\left[ |\hat{x}_\ell - x_\ell| > \alpha \max\{\Delta, \sigma\} \right] < 2 \exp\left( -\Omega\left( \alpha^2 k \right) \right) \ ,$$

*where* $\Delta = ||tail_b(x)||_2 / \sqrt{b}$.

**Discussion.** Theorem 3.3 generalizes the known tail bound on CountSketch, since if we let $\alpha = 1$ and $\sigma = 0$ we recover Theorem 3.1.

For $\varepsilon < 1$, if we choose suitable $\sigma = O(\sqrt{k \log(1/\delta)}/\varepsilon)$ the Private CountSketch satisfies $(\varepsilon, \delta)$-differential privacy. For noise level $\sigma \geq \Delta$ and deviation $\gamma < \sigma$ we then get:

$$\Pr[|\bar{x}_\ell - x_\ell| > \gamma] < 2 \exp\left( -\Omega\left( \gamma^2 \varepsilon^2 / \ln(1/\delta) \right) \right) \ .$$

Up to the hidden constant in the order-notation this is *identical* to the tail bound for the noise of the Gaussian distribution applied directly to $x$ in order to ensure $(\varepsilon, \delta)$-differential privacy.

### 3.3 Analysis of Private CountSketch

The message of our paper is that Gaussian noise composes very nicely with the CountSketches as analysed by (Minton & Price, 2014). All we use is that the CountSketch estimator is the median of symmetric variables.

For the median trick on indendent symmetric random variables $C_1, \ldots, C_k$, Minton & Price (2014, Lemma 3.3) show that for $\gamma > 0$ where $\Pr[|C_i| \leq \gamma] \geq p$ it holds that

$$\Pr[|\text{median}_{i \in [k]} C_i| > \gamma] \leq 2 \exp(-p^2 k/2) \ . \tag{3}$$

In our case, for $i = 1, \ldots, k$, we have a symmetric random variable $A_i$ (representing a simple estimator error before noise) to which we add Gaussian noise $B_i \sim N(0, \sigma^2)$. The estimator $C_i = A_i + B_i$ is clearly also symmetric. Basic properties of the Gaussian distribution implies that

    (a) if $\gamma \leq \sigma$ and $|A_i| \leq \sigma$ then $\Pr[|C_i| \leq \gamma] = \Omega(\gamma/\sigma)$.
    (b) if $\gamma \geq \sigma$ and $|A_i| \leq \gamma$ then $\Pr[|C_i| \leq \gamma] = \Omega(1)$.

In our concrete case (details below) we will have a furher property of $A_i$, namely

    (c) There exists $\Delta > 0$ such that for every $\alpha \in [0, 1]$, $\Pr[|A_i| \leq \alpha\Delta] = \Omega(\alpha)$.

**Lemma 3.4.** *Assuming (a), (b), and (c) then for every* $\alpha' \in [0, 1]$,

$$\Pr[|C_i| \leq \alpha' \max\{\Delta, \sigma\}] = \Omega(\alpha').$$

*Proof.* If $\alpha'\Delta \geq \sigma$ then using (b) and (c),

$$\begin{aligned} \Pr[|C_i| \leq \alpha' \max\{\Delta, \sigma\}] &= \Pr[|C_i| \leq \alpha'\Delta] \\ &\geq \Pr[|C_i| \leq \alpha'\Delta \mid |A_i| \leq \alpha'\Delta] \cdot \Pr[|A_i| \leq \alpha'\Delta] \\ &= \Omega(1) \cdot \Omega(\alpha') = \Omega(\alpha') \ . \end{aligned}$$

If $\alpha'\Delta \leq \sigma$ and $\Delta \geq \sigma$, we set $\alpha_1 = \sigma/\Delta$ and $\alpha_2 = \alpha'\Delta/\sigma$ such that $\alpha_1\alpha_2 = \alpha'$. Using (a) and (c) we get

$$
\begin{aligned}
\Pr[|C_i| \leq \alpha'\max\{\Delta,\sigma\}] &= \Pr[|C_i| \leq \alpha'\Delta] \\
&= \Pr[|C_i| \leq \alpha'\Delta \mid |A_i| \leq \sigma] \cdot \Pr[|A_i| \leq \sigma] \\
&= \Pr[|C_i| \leq \alpha_2\sigma \mid |A_i| \leq \sigma] \cdot \Pr[|A_i| \leq \alpha_1\Delta] \\
&= \Omega(\alpha_2) \cdot \Omega(\alpha_1) = \Omega(\alpha') \ .
\end{aligned}
$$

Finally, if $\alpha'\Delta < \sigma$ and $\Delta < \sigma$, we use (a) and (c) to get

$$
\begin{aligned}
\Pr[|C_i| \leq \alpha'\max\{\Delta,\sigma\}] &= \Pr[|C_i| \leq \alpha'\sigma] \\
&\geq \Pr[|C_i| \leq \alpha'\sigma \mid |A_i| \leq \Delta \leq \sigma] \cdot \Pr[|A_i| \leq \Delta] \\
&= \Omega(\alpha') \cdot \Omega(1) = \Omega(\alpha') \ .
\end{aligned}
$$

$\square$

We now return to the details of Private CountSketch. For a given $\ell \in [d]$, Private CountSketch finds $k$ simple estimates

$$
X_i = s_i(\ell)\left(CS(x)_{i,h_i(\ell)} + \nu_{i,h_i(\ell)}\right)
$$

of $x_\ell$ and returns the median, where $\nu_{i,h_i(\ell)} \sim N(0,\sigma^2)$ is Gaussian noise. Since the noise is symmetric, we get exactly the same distribution of $X_i$ if we compute it as

$$
X_i = s_i(\ell)\left(CS(x)_{i,h_i(\ell)}\right) + B_i
$$

where $B_i \sim N(0,\sigma^2)$. The point is that we can fix the variables in $s_i(\ell)\left(CS(x)_{i,h_i(\ell)}\right)$ first, and with the sign $s(i)$ fixed, $s_i(\ell)\nu_{i,h_i(\ell)} \sim N(0,\sigma^2)$.

In the above $s_i(\ell)(CS(x)_{i,h_i(\ell)})$ is the $i$th simple estimator of $x_\ell$ in the CountSketch. It has error

$$
A_i = s_i(\ell)(CS(x)_{i,h_i(\ell)}) - x_\ell
$$

so the error of Private CountSketch is distributed as $C_i = A_i + B_i$. For the error $A_i$ from Count-Sketch Minton & Price (2014) (proof of Theorem 4.1, using Corollary 3.2) proved that (c) is satisfied with $\Delta = \|\text{tail}_{b/2}(x)\|_2/\sqrt{b}$. Hence, by Lemma 3.4, $\Pr[|C_i| \leq \alpha\max\{\Delta,\sigma\}] = \Omega(\alpha)$ for every $\alpha \in [0,1]$. Now, by (3) for every $\alpha \in [0,1]$,

$$
\Pr[|\text{median}_{i\in[k]}C_i| > \alpha\max\{\Delta,\sigma\}] \leq 2\exp(-\Omega(\alpha^2 k)) \ .
$$

If $m$ is the index of the median $C_i$, then $X_m = x_\ell + C_m$ is the Private Countsketch estimator of $x_\ell$, so this completes the proof of Theorem 3.3.

We note that Theorem 3.3 could also be proved using the framework of (Minton & Price, 2014), exploiting that Gaussian variables have non-negative Fourier transform. However, this requires that the noise free error $A_i$ also has non-negative Fourier transform. Indeed this is the case for CountSketch as proved in (Minton & Price, 2014) if $b > 1$. However, our proof does not require $A_i$ to have non-negative Fourier transform, and this could prove useful in other contexts where Gaussian noise is added.

**Limitations.** Our error analysis relies on the assumption that the sign hash functions used are fully random. This assumption may be expensive to realize in practice, though it is always possible by storing an explicit, random list of all hash values (as done in our experiments). Alternatively, as observed in (Minton & Price, 2014) we could use the pseudorandom generator of Nisan (1990) to implement our hash functions, which requires space that exceeds the space for the sketch by a logarithmic factor. Furthermore, these hash functions can be shared among several CountSketches, reducing the space overhead.

Another issue to be aware of is that though CountSketch will be able to significantly compress sufficiently skewed or sparse input vectors, the size of a CountSketch with good accuracy can in general be larger than the size of the original vector.

### 3.4 Comparison to Other Private Sketches

It is instructive to compare Private CountSketch to other private sketches that have been studied in the literature, in particular Private Count-Min Sketch (Melis et al., 2016) and Count Mean Sketch (Apple Differential Privacy Team, 2017).

**Private Count-Min Sketch.** The Count-Min Sketch (Cormode & Muthukrishnan, 2005) is a well-known sketch that corresponds to CountSketch without the sign functions (or alternatively with constant $s_i(\ell) = 1$). Its estimator is similar to that of CountSketch except that it works by taking the *minimum* of the $k$ independent estimates obtained from the sketch. The minimum estimator has a *one-sided* error guarantee (never underestimates $x_\ell$) in terms of $||\text{tail}_{b/2}(x)||_1$ for vectors $x$ that have only nonnegative entries. Similar to CountSketch its failure probability decreases exponentially with the number $k$ of repetitions.

A private version of Count-Min Sketch, studied by Melis et al. (2016), works by adding independent (Laplace) noise to each entry of the sketch. This of course breaks the one-sided error guarantee. Another variant, with a non-negative binomial noise distribution (for integer valued vectors), was studied by Ghazi et al. (2021). In both cases the minimum estimator is biased and its variance grows with $k$. A third possibility would be Gaussian noise, but again the variance of the minimum estimator can be large (at least $O(k/\log k)$ for the minimum of $k$ Gaussians with variance $k$).

**Private Count-Mean Sketch.** We use Count-Mean Sketch to refer to CountSketch, where the estimator is the *mean* rather than the median. This sketch was studied by Apple Differential Privacy Team (2017) in the *local* model, where it was only used on 1-hot vectors with a single 1. One can ask if the mean estimator is useful in other models of differential privacy, but unfortunately it lacks the robustness properties of CountSketch. In particular, it gives error bounds in terms of the norm $||x||_2$ rather than $||\text{tail}_{b/2}(x)||_2$ which means that it is not robust against outliers or adversarial data.

**Local differential privacy** Zhou et al. (2022) consider differentially private encodings of $t$-sparse vectors with nonzero values in $[-1, 1]$. They use a CountSketch with a *single* repetition to encode a $t$-sparse vector, optionally apply a clipping step, and then add Laplace noise to ensure privacy. The main use case is the local differential privacy (LDP) model, where each noisy CountSketch is sent it to an aggrator that sums the estimates of $n$ users.

Kane & Nelson (2014) have shown that the norm of a CountSketch, $||Ax||_2$, with $k = O_\gamma(\log(1/\delta))$ repetitions is within a factor $1 \pm \gamma$ from $\sqrt{k}||x||_2$ with probability at least $1 - \delta$. In particular, the CountSketch of a $t$-sparse vector $x$ with nonzero values in $[-1, 1]$ has norm $||Ax||_2 < 2\sqrt{k}||x||_2 \leq 2\sqrt{kt}$ with probability $1 - \delta$. Applying clipping to ensure norm at most $2\sqrt{kt}$ and scaling the Gaussian noise by $2\sqrt{kt}$ our results imply an alternative LDP encoding of $t$-sparse vectors in $[-1, 1]^d$ that differs from the protocol of Zhou et al. (2022) by offering smaller error (in fact matching their lower bound) at the expense of larger communication complexity.

## 4 Experiments

We have conducted experiments to empirically investigate the properties of private CountSketch. Our main result ignores constant factors in the exponent of the bound on failure probability, but we see in experiments that these constants are very reasonable. All experiments can be recreated by running a Python script available on GitHub [1] (runs in ∼13 minutes on an M1 MacBook Pro). Density plots are smoothed using the Seaborn library's `kdeplot` function with default parameters.

### 4.1 Median of normals

As a warm-up we consider the setting in which a function $f(x)$ is released $k$ times independently, $r_1, \ldots, r_k \sim \mathcal{N}(f(x), k)$, for some integer $k$. The magnitude of the noise is chosen such that the privacy is bounded independently of $k$, e.g., if $f(x)$ has sensitivity 1 then $(r_1, \ldots, r_k)$ satisfies $1/2$-zero-concentrated differentially privacy (different parameters can be achieved by scaling the magnitude of the noise). Each release has noise of expected magnitude $\Theta(\sqrt{k})$, but the median

---

[1] `https://github.com/rasmus-pagh/private-countsketch/releases/tag/v1.0`

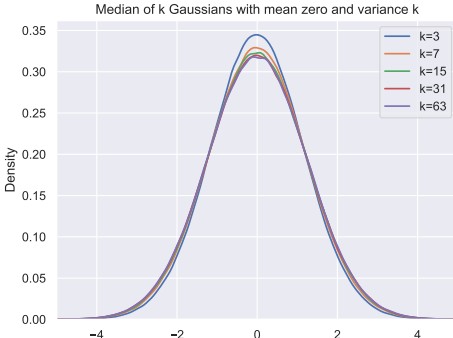

**Figure 1:** *It is well-known that the median of $k$ Gaussians with variance $k$ is itself subgaussian, with variance independent of $k$. This illustrates a "best case" for Private CountSketch, in which estimators $X_i$ (w/o noise) provide the exact answer.*

**Figure 2:** *Distribution of noise magnitude for Private CountSketch in the setting where Count-Sketch estimators have zero variance. The magnitude is only slightly higher than a standard Gaussian, achieving the same level of privacy.*

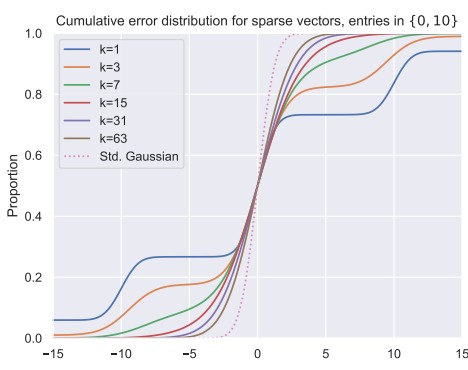

**Figure 3:** *Error distribution for Private Count-Sketch on sparse vectors for various number of repetitions, compared to making the output differentially private using a standard Gaussian, giving the same privacy guarantee.*

has noise that can be bounded independently of $k$, as illustrated in Figure 1. This gives a form of differentially private secret sharing, where $i$ of $k$ releases can be combined to yield an estimator with noise of magnitude $\Theta(\sqrt{k/i})$. Taking the mean value also gives a low-variance estimator, but mean values are sensitive to outliers and are not robust to adversarially corrupted data.

### 4.2 The zero-variance CountSketch setting

Next, we consider the idealized setting in which CountSketch itself does not have any error. This corresponds to letting the table size $b$ go to infinity, so we would hope to come close to the noise achieved by applying the Gaussian mechanism directly on the vector $x$. More generally, this is indicative of the situation in which the noise added to achieve differential privacy dominates the noise coming from randomness in CountSketch. Figure 2 shows the cumulative distribution function for the absolute value of the error obtained by Private CountSketch, for different values of $k$. (Of course, in this setting choosing $k = 1$ suffices to achieve a low failure probability, but a large value of $k$ is needed in general to ensure few estimation failures.) As can be seen, the noise distributions are quite close to that the standard Gaussian mechanism operating directly on $x$. To achieve a desired level of privacy, the noise in both cases has to be scaled appropriately, e.g., to achieve $(\varepsilon, \delta)$-differential privacy for $\varepsilon < 1$ it suffices to multiply the noise by a factor $\sqrt{2\ln(1/\delta)}/\varepsilon$.

### 4.3 Sparse vectors

Next we investigate the error distribution on $t$-sparse vectors, where at most $t$ entries are non-zero. It is well-known that a CountSketch with $k = O(\log d)$ repetitions each using space $b = O(t)$ is able

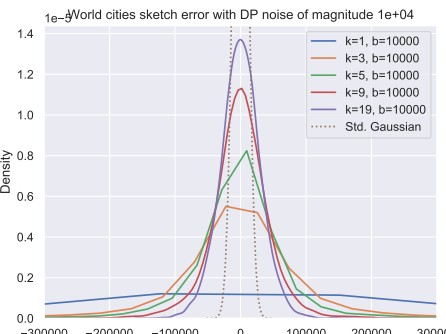
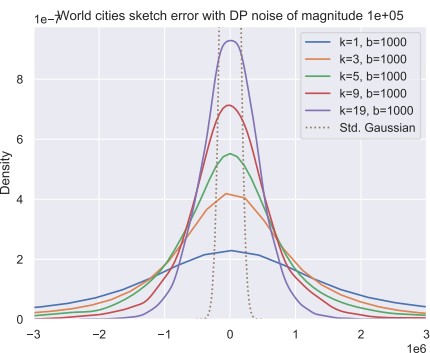

**Figure 4:** *Distribution of noise for Private CountSketch on the world cities dataset with table size $b = 10,000$ (left) and $b = 1000$ (right) for various number of repetitions $k$. The scale of the noise added for differential privacy is $\sigma = 10,000$ (left) and $\sigma = 100,000$ (right) so the sketch achieves strong group privacy. The Gaussian with standard deviation $\sigma$ is included for comparison.*

to reconstruct $t$-sparse vectors without any error, with high probability. Thus we expect the error from Private CountSketch, with a sufficiently large number of repetitions, to be similar to the error obtained by applying the Gaussian mechanism to the raw vectors. Figure 3 shows the cumulative error distribution for a Private CountSketch (event-level privacy) representing a $t$-sparse vector in which every nonzero entry has a value of 10, using $k$ repetitions with table size $b = t$. As can be seen, once the number of repetitions goes to 15 or more, the error distribution becomes comparable to that of the Gaussian mechanism applied directly to the sparse representation.

### 4.4 Real-world examples

**Population data with group privacy.**    To illustrate how Private CountSketch may be used on real-world data, we first consider the noise of a sketch of population counts in about 40,000 cities[2]. We can consider this as a very sparse, high-dimensional data set indexed by the names of cities (among the set of all possible strings). In this way, the sketch does not need to contain any direct information on the set of cities whose population counts are stored, though we will be able to infer such membership from auxiliary information on population (if this number is sufficiently high).

We aim for *group privacy* for large sets of people (see (Dwork & Roth, 2014) for a formal definition) by setting the target noise magnitude on estimates high, from $10^4$ to $10^5$. Intuitively, this hides the contribution to the sketch from any not too large group of people. The table size $b$ of the sketches is chosen to roughly balance the noise from the CountSketch itself and the Gaussian mechanism.

Figure 4 shows the figure with noise at scale $10^4$ (left) and at scale $10^5$ (right). As can be seen, the tails of the noise become noticably thinner as $k$ increased (of course at the cost of a factor $k$ in sketch size). Some of the sketches have size $kb$ larger than the original data set (the sparsity of the vector indexed by city names) so the point of sketching is not compression per se, though one would expect to see compression for larger and more skewed data sets.

**Market basket data.**    Finally, we consider representing two sparse histograms based on market basket data sets obtained from the FIMI data collection, `http://fimi.uantwerpen.be/data/`:

- The `kosarak` dataset (collected by Ferenc Bodon) contains click-stream data of a Hungarian on-line news portal, a set of click IDs per user session. We limit the size of each set to 100 clicks, resulting in 40148 distinct click IDs and 7264322 clicks in total. The high-dimensional vector considered represents the number of occurrences of each click ID.

- The `retail` dataset (collected by Tom Brijs) contains the shopping basket data from a Belgian retail store, a set of item IDs per customer. We limit the size of each set to 30 items, resulting in 16243 distinct item IDs and 888317 items in total. The high-dimensional vector considered represents the number of purchases of each item.

---

[2]Retrieved 2022-01-27 from `https://simplemaps.com/data/world-cities`

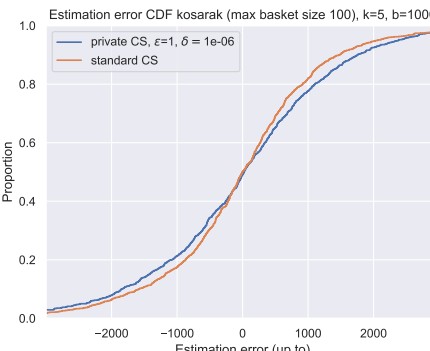
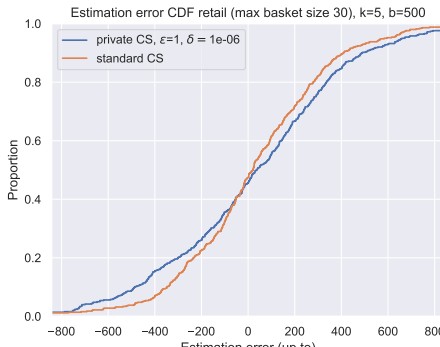

***Figure 5:*** *Cumulative distribution of noise for Private CountSketch on two market basket datasets,* `kosarak` *(left) and* `retail` *(right). Privacy is with respect to adding or removing a single basket, where baskets are truncated to a maximum size of 100 and 30, respectively.*

For both datasets we constructed CountSketches of size considerably smaller than the number of distinct IDs: $kb = 5000$ sketch entries for `kosarak` and $kb = 2500$ entries for `retail`. Figure 5 shows the empirical cumulative distribution functions of the error of CountSketch and Private CountSketch for the two datasets. For privacy parameters $\varepsilon = 1$ and $\delta = 10^{-6}$ (with respect to a single market basket) we see that the error of Private CountSketch is nearly as well concentrated as the error of CountSketch.

## 5    Conclusion

We have seen that CountSketch can be made differentially private at essentially the smallest conceivable cost: Coordinates of the sketched vector are estimated with error that is close to the maximum of the error from non-private CountSketch and the error necessary for differential privacy without sketching. A problem we leave is to obtain the same error bound with explicit, space-efficient classes of hash functions.

Since CountSketch can represent $t$-sparse vectors without error, for $k = O(\log d)$, this implies a new differentially private representation of such vectors that uses space $O(t \log d)$, is *dynamic* in the sense that the set of nonzero entries can be updated, and which keeps the level of noise close to the best possible in the non-sparse setting. This also implies that we can get good estimates of dot products between standard sparse vectors and vectors represented using Private CountSketch. Less clear is how to best estimate a dot product between two vectors given as Private CountSketches. This is related to work of Stausholm (2021) on differentially private Euclidean distance estimation, though one could hope for error guarantees in terms of the norm of the tails of the two vectors.

Finally, we note that Cohen et al. (2022) recently used differential privacy techniques in connection with CountSketch in order to achieve robustness against adaptive adversaries that attempt to find elements that appear to be heavy hitters but are in fact not. It is worth investigating whether Private Countsketch has similar robustness properties.

## Acknowledgments and Disclosure of Funding

We would like to thank the anonymous reviewers for their help with improving the exposition, and in particular with clarifying the relationship between our work and Minton & Price (2014). The authors are affiliated with Basic Algorithms Research Copenhagen (BARC), supported by the VILLUM Foundation grant 16582. Rasmus Pagh is supported by a Providentia, a Data Science Distinguished Investigator grant from Novo Nordisk Fonden.

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
