# OpenReview forum: "Improved Utility Analysis of Private CountSketch"
_NeurIPS.cc/2022/Conference — NeurIPS 2022 Accept_

### Official Review · Reviewer_x5cX · 2022-06-23

**Rating:** 7
**Confidence:** 4
**Soundness:** 4 excellent
**Presentation:** 3 good
**Contribution:** 3 good

**Summary:**

The paper present a novel analysis of CountSketch made differentially private using the Gaussian mechanism.  The analysis makes use of some novel/recent median-of-symmetric variables concentration analysis in a fruitful way.  The end result is that the performance (in terms of the noise added) matches what would occur with directly adding Guassian noise to the original vector to achieve differential privacy.  Experiments results are presented.

**Questions:**


Given my comments on the experiments above,  I am hoping you can provide some motivation/explanation of the experiments, and I would encourage you to more clearly motivate/explain them in the final version.

**Strengths And Weaknesses:**

Strengths:

The median analysis is very nice.
Seems like a worthwhile issue in differential privacy.

Weaknesses:

I didn't find the experiments and their explanations very clear.  For example, I couldn't understand what the high-variance CountSketch setting was supposed to demonstrate, or where this situation arises, or how these experimental results related to the theorems.  It's a setting that wasn't explained prior to the experimental section, does not seem to be standard (at least you don't reference other papers that have considered this setting), and it's not clear what the point is.  (A seemingly artificial setting where the private CountSketch can give better estimate than the standard CountSketch is not especially compelling.)  It could be my fault as the reader but I did not understand the purpose of that experiment.

Similarly I'm not clear on the importance/relevance of the sparse vector case -- it seems like you're just using it as a test, but it doesn't seem to be practically motivated, and is loosely related to the theoretical claims.

More compelling real-world datasets would also be useful.  In particular, you are using high noise for group privacy;  it feels like the setting with lower noise would still be worth examining and it would be useful to see it here as well.
[Overall, it felt like you could have added an appendix with more details/clarity on your experiments, and present more experiments.]

Neutral:

The authors note that their results require "full randomness" and that more limited randomness (e.g, k-wise independence) would not suffice.  They are up front about this assumption and the assumption itself is quite natural.

Originality:  Nice theoretical result.
Quality:  Same above.
Clarity:  Well written, except for the experimental section.
Significance:  Hard to say; if you believe people will actually use differential privacy, this seems a significant advance for applications using CountSketch, which I believe do exist.

---

> ### Author Response · Authors · 2022-07-31
> **Responser to Reviewer x5cX**
>
> Thanks for the thoughtful comments!
>
> Concerning the “high variance” case we looked for a clean setting demonstrating that the noise plus taking the median in fact can decrease the error relative to a plain CountSketch. This can happen, for example, when there are $k$ collisions with heavy elements, which is what this scenario was supposed to approximate. However, we admit that the error distribution is artificial and not one that would arise naturally. In the next revision of the paper we will instead seek to demonstrate this phenomenon on a real data set, with the sketch size set small enough that the the probability of having k collisions with heavy elements is nontrivial.
>
> We believe that the sparse vector case is well motivated. For example, sparse vectors appear in set reconciliation where we subtract two sketches to obtain a sketch of the difference between the vectors, which is sparse if the two vectors represent slightly out-of-sync versions of the same multiset. We refer to https://arxiv.org/abs/2112.03449 for more examples and discussion.
>
> Concerning further experiments on real datasets, an independent paper (https://arxiv.org/abs/2205.09873) recently made more extensive experiments than we did, including experiments on Private CountSketch. Their theoretical results are weaker than ours, but their experiments confirm the performance in practice.

---

### Official Review · Reviewer_CnGL · 2022-07-08

**Rating:** 6
**Confidence:** 4
**Soundness:** 3 good
**Presentation:** 3 good
**Contribution:** 2 fair

**Summary:**

CountSketch [Charikar-Chen-Farach-Colton 2004] is a popular data structure for maintaining a high-dimension vector $x\in \{0,1\}^d$ using a lower-dimensional representation $Ax\in \mathbb{R}^D$(think of $D\gg d$), while supporting single-entry estimations (i.e., given $i\in [d]$, return an additive estimation of $x_i$) and heavy-hitter queries (i.e., return a small list $L\subseteq [n]$ of indices, which includes all those $i$'s such that $|x_i|$ is large).

The standard CountSketch is parametrized by $(d,k,b)$. Intuitively, $b$ is the "table size" of CountSketch and $k$ is the number of "repetitions". It stores a representation of $x\in \mathbb{R}^d$ by a vector $Ax\in \mathbb{R}^{kb}$. Given a point query $i\in [d]$, it can return an estimation of $x_i$ (using the so-called median estimator) such that

$
\Pr\left[|x_i - \tilde{x}_i| > \frac{1}{\sqrt{b}} \| tail(x,b) \|_2 \right]\le e^{-\Omega(k)}.
$

The CountSketch can be made $(\varepsilon,\delta)$-differentially private by adding a Gaussian noise to the sketch vector: release $Ax+\nu$ where $\nu \sim N(0,\sigma^2)^{kb}$  and $\sigma^2 \approx \frac{2k}{\varepsilon^2} \ln(1.25/\delta)$. Intuitively, this distorts each entry of $Ax$ by a noise with variance $\sqrt{k}$. Therefore, one might expect the Private CountSketch can only support  point query with additive error $\Omega(\sqrt{k})$.

The authors argued that this is not the case by showing that, using the median estimator on top the Private CountSketch, one has

$
\Pr\left[ |x_i - \tilde{x}_i| > \frac{1}{\sqrt{b}} \| tail(x,b) \|_2 + \gamma \right] \le e^{-\Omega(\gamma^2\varepsilon^2/\ln(1/\delta))}.
$

Remarkably, this is independent of $k$ (the number of "repetitions" in the count sketch). (However, in order to recover the $e^{-\Omega(k)}$ upper bound on the failure probability, one must take $\gamma$ as $\Omega(\sqrt{k})$.)

The authors show extensive simulations to support their theoretical claims. In particular, note that there might be two sources of errors in the private CountSketch: (1) from the the inaccuracy of CountSketch itself (the $\frac{1}{\sqrt{b}}\|x_{tail(b)}\|$ term). (2) from the introduction of Gaussian noise. The authors consider different scenarios regarding which term of the errors is dominant and conduct experiments to see how the error scales with k. Overall, the finding is that as the number of repetitions $k$ increases, the error from the Gaussian noise dominates, and the PDF of the error "converges" to that of a Gaussian distribution.


**Questions:**

1. You mentioned that applying the Minton-Price technique in your setting directly yields weaker results. But it seems to me you can use the Minton-Price technique (https://arxiv.org/pdf/1207.5200.pdf) in a nearly black-box way to reproduce your main theorem. Let $X$ be an estimation of $x_\ell$ given by the (non-private) CountSketch. Let $\Delta = 1/\sqrt{b} ||x_{tail(b)} ||$. Let $g\sim N(0,\sigma^2)$. Essentially, your proof wants to lower bound $\Pr[X + g \in [x_{\ell} - \Delta - \gamma, x_{\ell} + \Delta + \gamma]]$ by $\Omega(\gamma/\sigma)$. You achieve this by showing that $\Pr[X \in [x_{\ell} - \Delta, x_{\ell} + \Delta]] \ge \Omega(1)$ and $\Pr[|g|\le \gamma] \ge \Omega(\gamma/\sigma)$. However, if we consider $Y := X + g - x_{\ell}$, it is easy to see that $Y$ (1) has varaince $\le \sigma^2 + \Delta^2$, (2) is symmetric around zero, and (3) has non-negative Fourier transform. Using Lemma 3.1 in [ https://arxiv.org/pdf/1207.5200.pdf ] should allow you to recover (and slightly improve if $\Delta \ll \sigma$) your result. (Please correct me if I make a mistake at some step.)
2. In your discussion at Line 171: it should be $|x_{tail(b)}|/\sqrt{b} < \gamma < \sigma$ (the submission stated $\gamma > \max(\sigma, |x_{tail(b)}|/\sqrt{b})$?
2. Minor typo: the equation below Line 206: $x_i + \Delta$ ==> $x_\ell + \Delta$.

**Ethics Review Area:**

["I don’t know"]

**Limitations:**

See the "weakness" and "questions" sections.

**Strengths And Weaknesses:**

Strengths:

- The observation that the median estimator is "compatible" with Gaussian noise is nice and elegant. Although this seems not completely novel (see the "weakness" and my question below), this paper manages to apply this observation in the context of sketching + privacy, which allows privatizing CountSketch in a simple and standard way, while still achieving a good utility guarantee.
- The experiments are thorough and illustrative, which I appreciate. Although I did not try to reproduce the simulations myself, the presentation looks sound and clear.
- Discussions about related work and comparisons with other private sketches are well-written and informative.

Weakness:

- The main technical message of this work is that "taking the median of several i.i.d. random variables, you can expect the result deviation (aka. error) to be much smaller than the standard deviation of the individual random variables". This appears not to be a completely novel observation. In particular, it appears in a previous work [Minton and Price, SODA 2014. see also https://arxiv.org/abs/1207.5200 ]. The current paper did mention the Minton-Price result, but the discussion and comparison seem not quite fair (see my question below).
- The theoretical analysis needs to assume that the hash functions for CountSketch are truly random. Storing a truly-random hash function would require a space as large as the dimension of the input vector. To compare, the standard analysis for CountSketch (as in [Charikar-Chen-Farach-Colton 2004]) only needs pairwise-independent hash functions, which can be stored using a space that is logarithmic in the dimension of x.

My overall opinion is that (1) the technical result in this paper is not completely novel, given the previous Minton-Price result. But the current paper does find a new application of the Minton-Price idea to an important question (i.e., privatizing CountSketch). (2) The presentation is good, the experiments are thorough, and the discussions about related works are informative.

In summary, I would recommend a weak accept.

---

> ### Author Response · Authors · 2022-07-31
> **Response to Reviewer CnGL**
>
> Thanks for your insightful review! It prompted us to look closer at the techniques in the Minton-Price paper (which we were not familiar with at the time of submission), and we agree that a discussion of the relationship should be added. For example, we found that their Lemma 3.3 corresponds to an observation made in our proof, and we should of course acknowledge this.
>
> Our key contribution is the realization of our Lemma 3.1 which states that the median-trick on symmetric estimators composes perfectly with symmetric random noise (e.g., added for privacy). This is a simple quotable lemma that could easily be applied in other contexts. . Though Private CountSketch has been studied in several previous papers (and recently in https://arxiv.org/abs/2205.09873, independent of our paper) it was not known before that the noise would behave so well.
>
> Concerning question 1: This is a great suggestion. It seems that it is possible to obtain error probability $2\exp(-\Omega(k\, \min\{1, \gamma/\Delta)\}^2 ))$ where $\Delta=\delta+||\text{tail}_{b/2}(x)||_2 / \sqrt{b}$ in this way (for any $\gamma > 0$). Is that the type of bound you had in mind?
>
> Mathematically it is true that this can be derived from Minton-Price by someone sufficiently familiar with their use of the Fourier transform. Here we present a simple self-contained combinatorial proof that is less than one page. In fact, it turns out that we do not need to use the specific techniques in Minton-Price to get such a result: We can show that *any* tail bound for CountSketch (with fully random hash functions) composes perfectly with Gaussian noise. Thus, we can get this stronger result using Minton-Price as a black box.
>
> Thanks for catching the typos in questions 2 and 3!

---

> > ### Author Response · Authors · 2022-08-01
> > **A small added comment**
> >
> > I just wanted to add a small comment to our previous comment; namely that our reduction applies black-box to any tail bound on symmetric estimators, e.g., potentially, it could be something different from CountSketch, and the estimators would not necessarily have to have non-negative Fourier transform (although the use of non-negative Fourier transform in Minton-Price is very elegant). Once again, thank you so much for your insight. It was fun to study Minton-Pride closer, and very relevant to also consider the smaller errors they consider in our reduction.

---

> > > ### Comment · Reviewer_CnGL · 2022-08-07
> > > **Thanks for your comments**
> > >
> > > I am glad to see that the authors agree with my points. While I agree that applying the median trick to give an improved analysis for private CountSketch (and potentially other symmetric sketches) is new and elegant, I still would like to encourage the authors to add a discussion and comparison with the Minton-Price result. In my opinion, the key contribution of Minton-Price consists of two parts. (1) they observed that for a symmetric random variable X, lower-bounding Pr[ X >= c ] by Omega(c) automatically gives the improved deviation bound for the median many X_i's. (2) they introduced some new Fourier analytic techniques to implement this observation. The first point is also your key technical point. In your context, given this observation, lower-bounding the deviation of Gaussian noise by $\Omega(\gamma/\sigma)$ is somewhat immediate (at least to me).
> > >
> > > Thank you again for answering my concerns. I would like to keep my current score.

---

> > > > ### Author Response · Authors · 2022-08-07
> > > > **I think we agree what has to be done in relation to Minton-Price**
> > > >
> > > > Dear Reviewer (CnGL),
> > > >
> > > > I think we now agree on the facts concerning the relation between our work on the nice work of Minton-Price. We will add a discussion and comparison as you suggests. Once again, thank you for your comments.

---

> > > > > ### Author Response · Authors · 2022-08-08
> > > > > **One more comment to Reviewer (CnGL)**
> > > > >
> > > > > Dear Reviewer (CnGL),
> > > > >
> > > > > Just want to add that our main contribution is to discover the "perfect" match between Gaussian noise and the median of the symmetric estimators as in CountSketch, e.g., previous authors had looked at privacy for CountMinSketch, and even those who considered privacy of CountSketch had not noticed what we saw.
> > > > >
> > > > > The analysis itself is not hard. Our self-contained analysis was a bit more than a page, and it will be simplified with reference to Minton-Price as you suggest. I am stating this because our main contribution is *not* some individual step in the analysis, so when our equation (3) can now be obtained directly as Lemma 3.3 in Minton-Price, then it doesn't change much (except that we save some lines). Likewise, I agree when you say " lower-bounding the deviation of Gaussian noise is somewhat immediate". As stated, the main contribution is the discovery of the match. When first you are told to analyze this match, then the rest is not hard (even easier if you know Minton-Price). The connections to Minton-Price will be made clear giving them full and well-deserved credit.

---

### Official Review · Reviewer_o9ww · 2022-07-12

**Rating:** 7
**Confidence:** 3
**Soundness:** 3 good
**Presentation:** 4 excellent
**Contribution:** 3 good

**Summary:**

The paper considers CountSketch that is made private using Gaussian mechanism. They show that the privacy-utility trade-off is essentially tight.

**Questions:**

No questions

**Strengths And Weaknesses:**

The paper gives a new analysis of private count sketch, where privacy is achieved using Gaussian mechanism. They show that median estimator has better bounds than mean estimator.

The proof is correct and the paper does improves the general understanding of the area. I recommend accepting the paper.

epsilon should be less than one in Lemma 3.2 .

---

> ### Author Response · Authors · 2022-07-31
> **Response to Reviewer o9ww**
>
> Thanks for your comments. Indeed, we forgot to include the assumption that $\varepsilon < 1$ in Lemma 3.2, which is required for $(\varepsilon,\delta)$-DP with the stated parameters. (Note that the assumption is made for simplicity and is not essential, the Gaussian mechanism with $\varepsilon \geq 1$ is well-understood.)

---

### Meta-Review · Area_Chair_xMmH · 2022-08-23

**Recommendation:** Accept
**Confidence:** Certain

**Metareview:**

The paper presents an improved analysis for a differentially private variant of CountSketch, leveraging concentration  bounds regarding the median of symmetric variables. The reviewers found the new analysis interesting and the new results significant.

**Award:**

No

---

### Decision · Program_Chairs · 2022-09-14

Accept